# Emerging dynamics from high-resolution spatial numerical epidemics

Olivier Thomine[1†]*, Samuel Alizon[2], Corentin Boennec[2], Marc Barthelemy[3], Mircea Sofonea[2]

[1]LIS UMR 7020 CNRS, Aix Marseille University, Marseille, France; [2]MIVEGEC, Université de Montpellier, CNRS, IRD, Montpellier, France; [3]Institut de Physique Théorique, CEA, Saclay, France

**Abstract** Simulating nationwide realistic individual movements with a detailed geographical structure can help optimise public health policies. However, existing tools have limited resolution or can only account for a limited number of agents. We introduce Epidemap, a new framework that can capture the daily movement of more than 60 million people in a country at a building-level resolution in a realistic and computationally efficient way. By applying it to the case of an infectious disease spreading in France, we uncover hitherto neglected effects, such as the emergence of two distinct peaks in the daily number of cases or the importance of local density in the timing of arrival of the epidemic. Finally, we show that the importance of super-spreading events strongly varies over time.

## Introduction

Mathematical modelling is a powerful tool to describe infectious disease epidemics, for example when combined to statistical modelling, and also to understand ongoing processes (*Kermack and McKendrick, 1927*; *Keeling and Rohani, 2008*). The recent COVID-19 pandemic has put in the spotlights the importance of mathematical epidemiology models to elaborate intervention strategies (*Adam, 2020*). These models face important challenges such as stochasticity, spatial structure, or individual heterogeneity. In the initial stages of an outbreak, the effect of spatial structure is minimal because transmission chains are largely independent, but individual heterogeneity and stochasticity have major effects (*Trapman et al., 2016*; *Britton and Scalia Tomba, 2019*). As the epidemic unfolds and host become immune, accounting for the exact shape of local contact networks matters increasingly (*Keeling and Rohani, 2008*). Several approaches have been developed in epidemiological models to capture spatial structures, for example meta-populations (*Grenfell and Harwood, 1997*), moment equations (*Lion, 2016*), or contact networks (*Pellis et al., 2015*). However, one of their limitation is that they simplify the geographical structure (sometimes ignoring it completely), which can lead to overlooking emerging patterns and complicate the practical implementation of local policies.

Agent-based simulation (ABS), where individuals are modelled explicitly, represent a seducing option to achieve a high degree of realism, from a biological and environmental point of view (*Abar et al., 2017*), but their routine use in public health faces three major limitations. First, ABS are very computationally demanding (*Eubank et al., 2004*), which restrains the total number of agents that can be simulated. The second limitation comes from the model dimensionality and the introduction of numerous parameters, many of which are poorly informed and set in an *ad hoc* manner, to capture individual heterogeneity. A third limit resides in the way the geographic structure is implemented into the simulation.

The recent SARS-CoV-2 pandemic has led to the creation or the re-implementation of ABS that alleviate some of the limitations. For instance, some of these simulations were used to introduce

**\*For correspondence:**
olivier.thomine@protonmail.com

**Present address:** †CNRS-LIS, Marseille, France

**Competing interests:** The authors declare that no competing interests exist.

biological details into the model that would require numerous equations in an analytical approach (see e.g. *Kerr et al., 2021*). Other ABS introduced some spatial structure to tackle generic questions, such as the impact of individual movement on epidemic spread. Given the nature of the questions asked and also given the additional types of structures in the model, for instance in terms of individual ages of households, these simulations typically simplified the spatial structure using contact networks (see e.g. *Hinch et al., 2021*, *Kerr et al., 2021* at a city level, or *Aleta et al., 2020* for unstructured network regenerated by contact tracing). To our knowledge, few ABS feature high-resolution geographical scale. There are exceptions and, for instance, *Smieszek et al., 2011* analyse the spread of influenza virus in Switzerland using an ABS using a grid with 500x500m resolution. *Rockett et al., 2020* perform a similar analysis in Australia using the 2310 level 2 statistical areas Also, the recent work by The GAMA platform allows one to combine a detailed epidemiological model with a high geographical resolution (*Taillandier et al., 2019*). In general, achieving a high degree of geographical realism for a large number of agents is an open challenge in epidemiology.

Here, we introduce Epidemap, a novel numerically efficient agent-based method that addresses many limitations of current ABS platforms. In particular, it can simulate infectious diseases epidemic scenarios at the scale of a whole country by combining high-resolution geographical structure, demographic information, and mobility statistics. Practically, this is achieved by using high-performance computing (HPC) techniques, building-level spatial data from the OpenStreetMap (OSM) project (*Haklay and Weber, 2008*), and sophisticated mobility models (*Barbosa et al., 2018*). Overall, in addition to the number and age of the hosts (which is informed by the demography) and the initial conditions (number of infected hosts at $t = 0$), these simulations only require seven parameters (see Table 2).

To illustrate the power and flexibility of the platform, we study the transmission dynamics of an (uncontrolled) epidemic of respiratory infections in France. The biological features of the epidemiological model originate from a discrete-time SARS-CoV-2 transmission model parameterised with national hospital data (*Sofonea et al., 2021*). Given the general focus of the study, we voluntarily focus more on the transmission dynamics than on the clinical dynamics but both are implemented. We analyse the output of 100 stochastic simulations from epidemic emergence to extinction, that is, approximately 1 year. A typical simulation generates daily mobility patterns for 66 million individuals at the scale of buildings and lasts less than 2 hr. Further details about the simulation specifications and comparisons with existing platforms can be found in the Materials and methods.

In the simulations, which are summarised in *Figure 1* and detailed in the Materials and methods, each individual is assigned to a residency building and can visit two other buildings every day. These buildings are chosen at random based on a distance kernel (see the Materials and methods). If more than one individual visits the building the same day, transmission can occur. The life-history of the infection is parameterised using data from the COVID-19 epidemic in France (*Sofonea et al., 2021*). Note that most of the parameters are required to simulate Intensive Care Unit (ICU) bed occupancy dynamics, which do not affect the transmission dynamics. Given the general scope of this study, we assume that recovered individuals have perfect immunity for the duration of the simulation (i.e. a classical Susceptible Infected Recovered (SIR) epidemiological formalism) and that all simulations are initialised with 15 infected individuals in Paris to avoid premature random epidemic extinction. Similarly, although the simulation tracks the age of the individuals, which is distributed geographically according to national demographics data (*INSEE, 2020*), following a parsimony principle, we assume that it does not affect the mobility or the transmission model (but age does affect the clinical model).

Epidemap simulations do not attempt to reconstruct a past epidemic. They are not conceived either to perform statistical parameter inference. Their main goal is to simulate realistic scenarios to better understand how mobility patterns, geographical structure, and infectious disease biology interact to shape epidemic spread and to optimise public health responses.

## Results

*Figure 2* shows the output of the dynamics at the national level for 100 stochastic simulations. For optimal readability, the dates are aligned based on the day where ICU-occupancy reaches a value of 700. With our minimal parameterisation, we see that the basic reproduction number, which is denoted ($R_0$) and corresponds to the average number of secondary infections caused by an infected

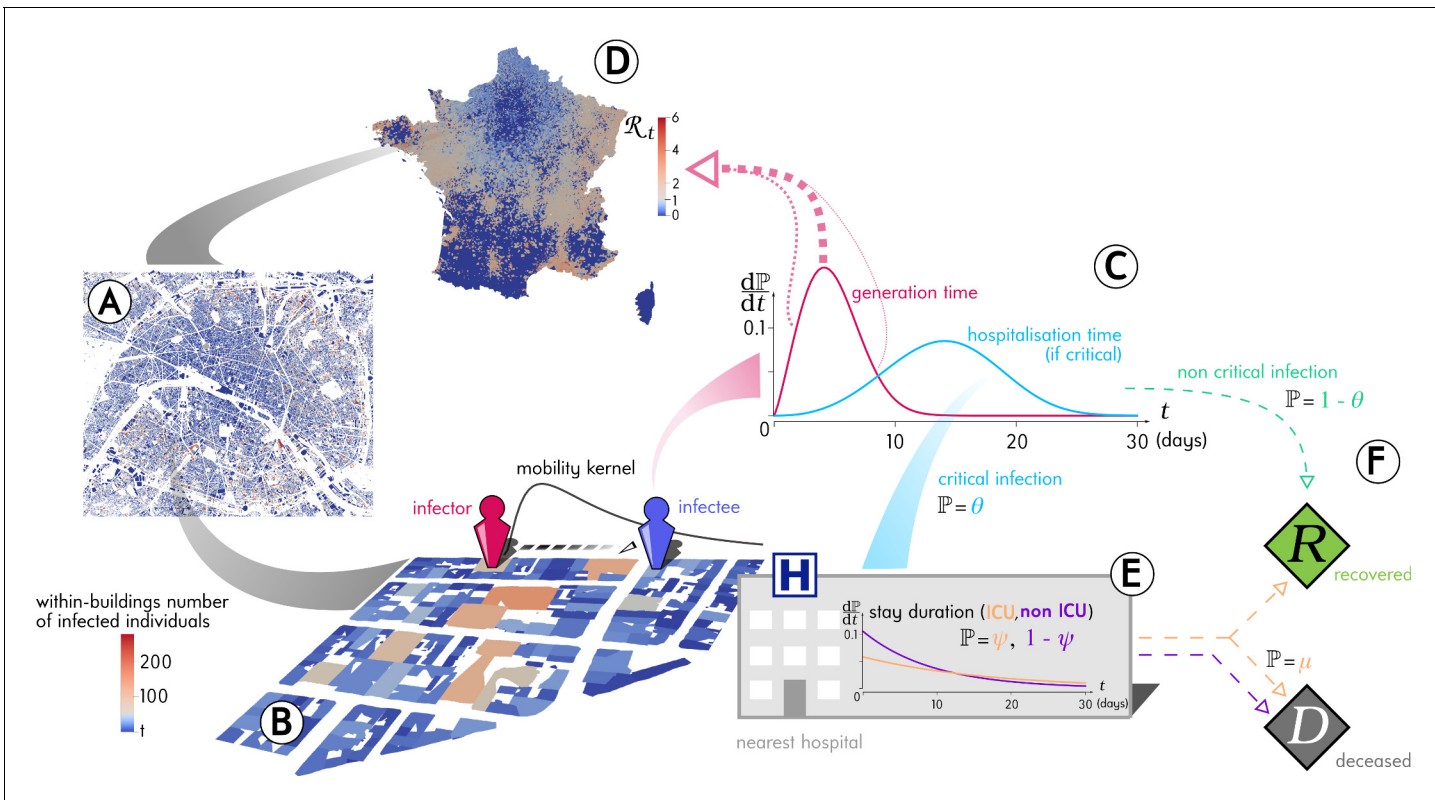

**Figure 1.** Outline of the Epidemap simulation framework. (**A**) The 66 millions inhabitants of metropolitan France are explicitly mapped to housing buildings following cartographic and demographic data. (**B**) At each time point of the simulation, the number of infected individuals in each building of the country is recorded, as well as the time past since each got infected (the panel shows the Paris area). (**C**) Every day, individuals can randomly move from their home to other buildings according to a mobility kernel and meet other people. If an infected individual encounters a susceptible host, a transmission event can occur. (**D**) The contagiousness of a infected individual varies depending on the time since infection. (**E**) A small fraction $\theta$ of infected individuals develop a critical form of the disease that requires their hospitalisation to the nearest facility. Clinical dynamics can be assumed not to affect transmission dynamics because more than 95% of the secondary transmission events occur before hospital admission.

individual (*Anderson and May, 1991*), is of the order of 3, which is consistent with estimates for the French epidemic (*Salje et al., 2020*; *Sofonea et al., 2021*). Note that this value is here computed directly at the individual level, by counting how many infections an individual causes. Furthermore, the daily estimates for the temporal reproduction number calculated in the same manner are very similar to those estimated using the daily case incidence data (dashed blue curve) and the method from *Wallinga and Lipsitch, 2007*. These uncontrolled epidemics last 308 days (95% confidence interval (CI): [286;345] days) and the final total epidemic size is $q = 61.1\%$ (95% CI [60.1%;61.9%]) of the initial susceptible population. As described by earlier studies (*Keeling, 1999*), this proportion is lower than the prediction from a mean-field model that is given by the well-known equation $qR_0 + \log(1-q) = 0$ from *Kermack and McKendrick, 1927*, which yields 93% for $R_0 = 3$. This shows that geographical structure greatly impacts the unfolding of the epidemic.

The national prevalence data uncovers a bimodal structure of the epidemic, which can be understood by moving to the regional scale (*Figure 2b*). The first peak corresponds to the spread in the region where the outbreak emerges (here the Ile-de-France), whereas the second corresponds to the sum of the epidemic peaks in the other regions. This bimodal structure is particularly pronounced because of the high population density in the region of origin of the outbreak (Ile-de-France), but it is a direct consequence of the detailed geographical information in the simulation platform. As expected, in a well-mixed setting the parameters used for the transmission model yield a single peak (*Figure 2—figure supplement 1*).

The resolution of the Epidemap simulations allows us to perform analyses at the district level (see *Videos 1* and *2*). In *Figure 3a*, we show that the date of onset of the epidemic in an area strongly depends on its distance from the origin (here assumed to be Paris). Furthermore, there is an

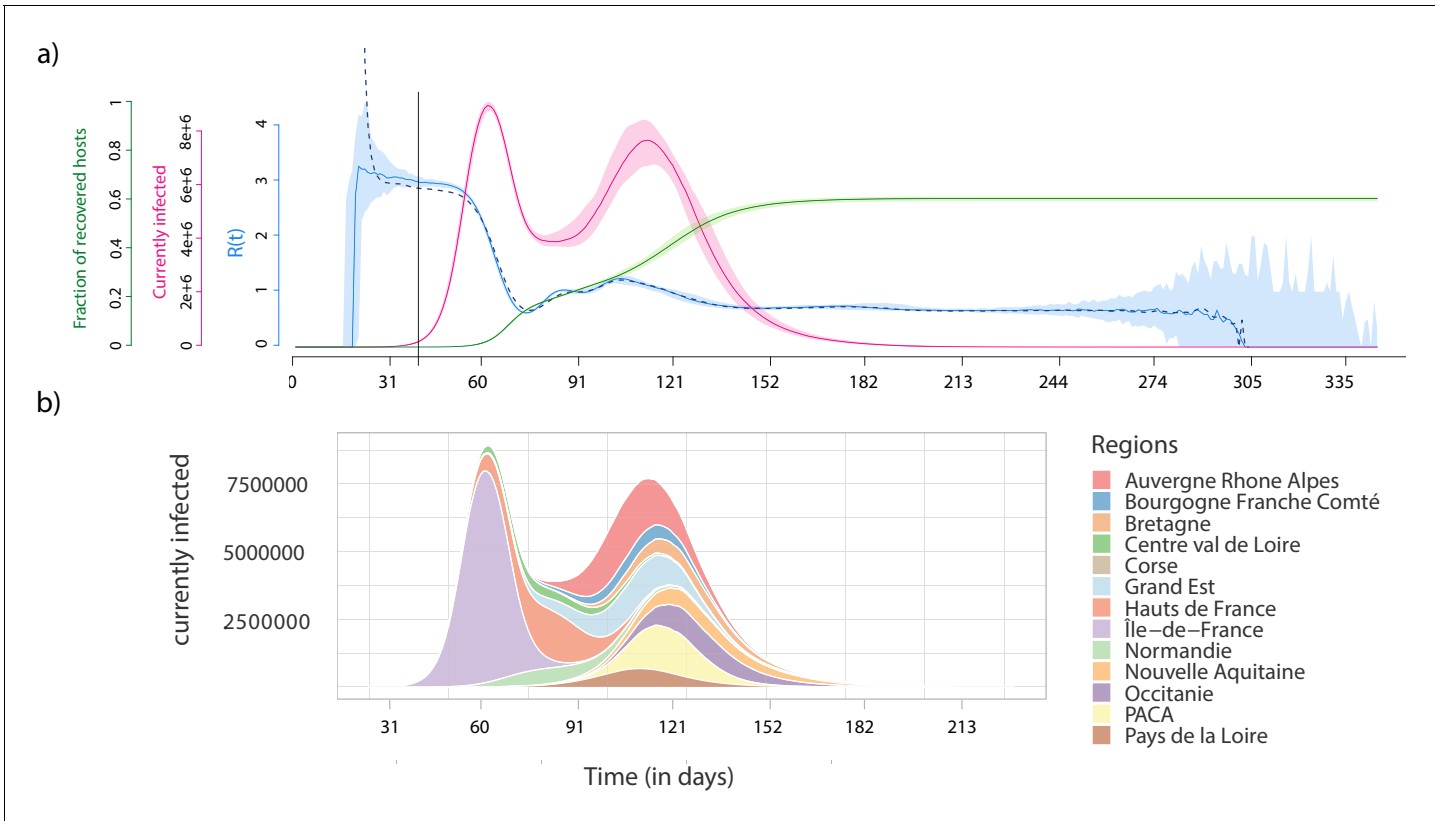

**Figure 2.** Epidemiological dynamics at the national (**a**) and regional (**b**) level. (**a**) The daily prevalence (the number of infected individuals) is in red, the temporal reproduction number ($R_t$) in blue, and the cumulative number of recovered individuals in green. Shaded areas show the 95% sample quantiles of 100 stochastic simulations. The dashed line shows the median $R_t$ calculated on the case incidence data. (**b**) Each colour shows the prevalence in a French region.

The online version of this article includes the following figure supplement(s) for figure 2:

**Figure supplement 1.** Mean field epidemiological dynamics.

additional effect of density such that denser areas are infected first. Interestingly, at the departmental level, these trends are not significant, further showing the importance of a fine-grain simulation level. Furthermore, the total proportion of inhabitants of a district who have been infected at the end of the epidemic strongly increases with density. The pandemic propagation velocity also increases with time. This can be explained by the fact that, when incidence is high, long-distance dispersion events are not rare anymore, which biases the mean distance of contamination towards higher values. For densely populated districts (*Figure 3b*), this proportion converges towards the mean-field prediction from well-mixed models mentioned above (*Kermack and McKendrick, 1927*). However, if the population density is low, this proportion is more variable showing the limit of classical assumptions.

The individual-based nature of our simulations allows us to follow transmission chains (*Video 3*), which has direct applications. For instance, we can count, at the end of each infection, how many secondary infections were caused. The distribution of these individual reproduction numbers is particularly important in the context of emerging epidemics because the more disperse, the more the spread relies on super-spreading events (*Lloyd-Smith et al., 2005*). Early in the epidemic, the distribution is tightly centred around the $R_0$ value (*Figure 4a*). As the epidemic unfolds, the distribution changes with a mode that decreases towards 0, and a wider dispersal. This pattern can be formalised by assuming that the distribution of individual reproduction numbers follows a negative binomial distribution (*Lloyd-Smith et al., 2005*). In *Figure 4b*, we show that the mean of this distribution (in blue) follows the pattern estimated in *Figure 2a*. The dispersal parameter ($k$) indicates that super-spreading events reach a peak at the end of the first national epidemic wave. During the end of the

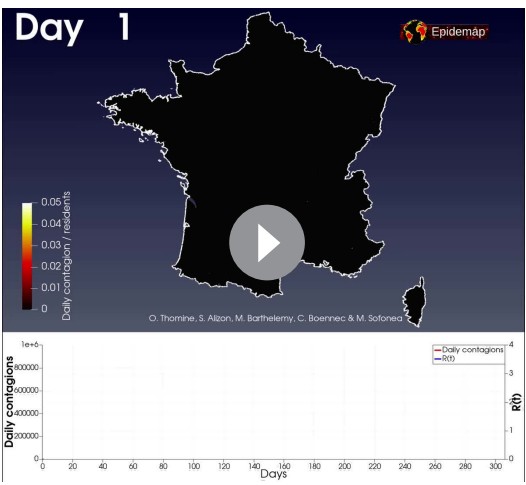

**Video 1.** Density of residents in ICU throughout an uncontrolled epidemic. The top panel shows the value at a fine geographical scale (French 'canton') and the bottom panel shows the total density at the national level along with the corresponding temporal reproduction number ($R_t$).

https://elifesciences.org/articles/71417#video1

epidemic, stochasticity is strong but there is a general trend towards a decrease in super-spreading events. Note that in these simulations we do not introduce host heterogeneity, which means we only capture the dimension of super-spreading that originates from spatial heterogeneity. This is already sufficient to show that studies attempting to quantify the importance of superspreading events should account for the stage of the epidemic they analyse.

Finally, we focused so far on the infection spread but by adding a clinical part to the infection model, and therefore additional parameters (Table 3), we can also capture the hospital epidemic wave dynamics. For instance, *Video 1* shows the detailed geographical dynamics of the density of residents in ICU. As expected in the case of SARS-CoV-2, where clinical dynamics have little effect on transmission dynamics due to the life-history of the infection (see the Materials and methods), these dynamics closely follow that of infection cases.

## Discussion

Simulating the daily activity of millions of individuals at a build-level resolution with a realistic mobility model significantly improves our understanding of how epidemics unfold. Early stages appear to be consistent with stochastic and deterministic mean-field models. However, once local saturation effects cannot be neglected anymore, Epidemap reveals striking patterns with an unprecedented resolution. First, a two-waves epidemic pattern emerges at the French national level, which is largely driven by temporally shifted dynamics at the regional level. Second, we find that districts are affected by the epidemic depending on how far they are from the epicentre, but also depending on their density. The latter effect is absent at the departmental level, which illustrates the added value of a detailed geographical structure. Furthermore, as expected (*Keeling, 1999*), the simplistic estimate of the final epidemic size as a function of $R_0$ does not apply at the national level. Conversely, this estimate does yield relevant results at the district level if the density is sufficiently high. Finally, being able to perform individual follow-ups allow us to see that superspreading events become increasingly important as the epidemic unfolds, but that their role decreases as the importance of stochastic processes increase again at the end of the epidemic wave. In general, many insights can also the gained from the ability to follow individual trajectories and transmission chains (*Video 3*), and superspreading events.

Replacing these results in the broader context of infectious disease epidemiological modelling helps to better identify the originality of the approach. For instance, the link we identify

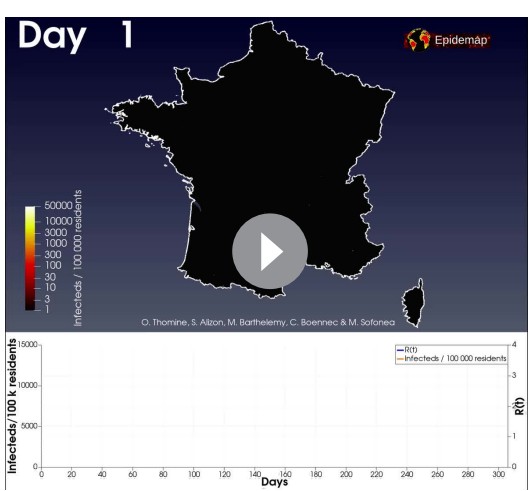

**Video 2.** Density of infected people / 100 k residents for an uncontrolled epidemic. The top panel shows the value at a fine geographical scale (French 'canton') and the bottom panel shows the total density at the national level along with the corresponding temporal reproduction number (R(t)).

https://elifesciences.org/articles/71417#video2

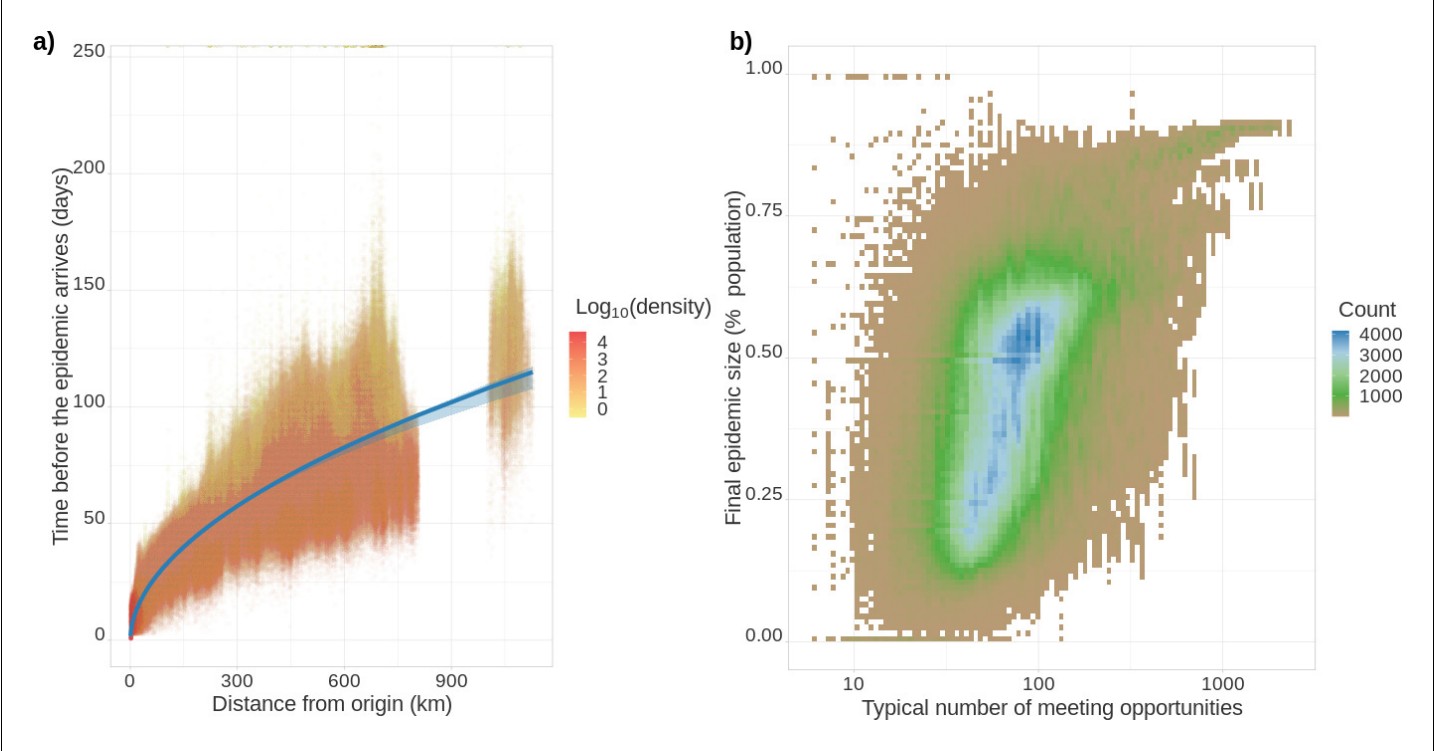

**Figure 3.** Epidemic arrival date and size at the district level. (a) Effect of the distance from the epicentre on the number of days until the epidemic begins in a district. The colour indicates the population density in the district (number of inhabitants divided by the district's surface). (b) District final epidemic size as a function of the characteristic distance between two individuals normalised by the average dispersal distance. The latter is computed as $2E[X]\sqrt{\text{density}}$, where $X$ is the log-normal distribution of daily individual covered distance. Both panels show the value for 35,234 French districts and 100 stochastic simulations.

between the final size of the epidemic in a district and its population density is not reported in the other ABS we discussed here, but appears quite strongly in SARS-CoV-2 data (*Smith et al., 2021*). Similarly, superspreading events are known to be an important target for public policies because they increase the risk of stochastic extinction but also to fuel the speed of spread of epidemics that do emerge (*Lloyd-Smith et al., 2005*). However, although field studies find that their importance may vary over the course of an epidemic (*Lau et al., 2017*), we are not aware of an in-depth analysis of these emerging trends using ABS. Note that since Epidemap can store transmission chains, which resemble infection phylogenies, we could further investigate the possibility to detect superspreading events using various kinds of data (*Alizon, 2021*).

Being able to perform such detailed simulations for so many agents at a national scale has to be traded-off against some simplifying assumptions. The major one is that individual movements are based on a distance kernel centered in their residency home. As shown by earlier work on influenza dynamics, although this assumption is relevant for France, it might be too simplistic for countries like the United States, where air traffic represents a greater proportion

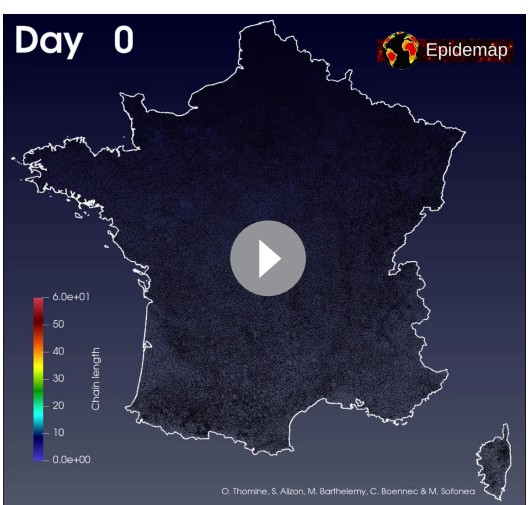

**Video 3.** Complete transmission chains colored by chain length of an uncontrolled epidemic.
https://elifesciences.org/articles/71417#video3

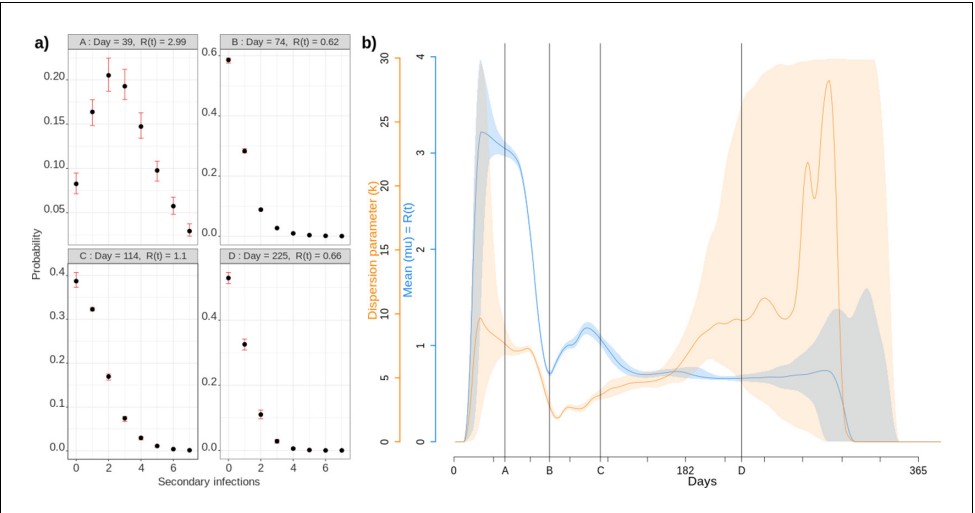

**Figure 4.** Individual reproduction number dynamics. (**a**) Distribution obtained over the whole population on 4 different days post outbreak. (**b**) Daily variation of the mean (in blue) and dispersion (in orange) of the distribution of individual reproduction numbers, which is assumed to follow a Negative Binomial distribution (*Lloyd-Smith et al., 2005*). Shaded areas show the 95% CI.

of the travels (*Crépey and Barthélemy, 2007*). A second limitation is that, because of the lack of relevant data, we assumed all individuals to have the same average behaviour, *e.g.* in terms of the number of buildings visited per day. This assumption was motivated by the will to develop a parsimonious study and not by a constraint of the Epidemap formalism. In fact, as mentioned in the Materials and methods, we already follow the age or the infectious status of the agents and could easily have mobility that depend on these parameters.

Focusing more specifically on the importance of individual age, there are several ways in which accounting for this data in the simulations could be of interest. First, we could investigate how patterns observed for some 'childhood' infectious diseases emerge in the simulations. Indeed, one possibility is that these are only due to children having specific mobility patterns, which could be modelled here by imposing that instead of visiting two buildings at random, they only visit the nearest school. Another possibility could be that explaining childhood infectious disease dynamics require children-specific patterns in the infection model, that is susceptibility to the infection and/or contagiousness. Finally, both could also be needed. By separating the mobility model and the infection model, and by introducing a high-resolution and realistic mobility model, Epidemap could yield original insights into such observed epidemiological patterns.

These results have immediate applications for public health. For instance, they allow authorities to derive a risk factor per district to help control an epidemic and prevent outbreaks. From an even more applied perspective, Epidemap can readily simulation hospitalisations by sending an individual to the nearest hospital, thereby allowing to anticipate the saturation of ICU at a detailed geographical level. In the context of the SARS-CoV-2 pandemic, this simulation platform can also be instrumental to optimise vaccine coverage but also compare the effect of specific Non-Pharmaceutical Intervention (NPI) (e.g. mask-wearing vs. stay-at-home requirements).

We focused here on respiratory infection similar to SARS-CoV-2 but an asset of this platform is its versatility. We already mentioned how it could be modified to investigate childhood respiratory infections. Simulating sexually transmitted infections could prove to be challenging with such a framework because these require to model partnerships and a fine-grain geographical resolution seems less crucial (*Althaus et al., 2012*). However, a more feasible extension could be the study of vector-borne diseases. Indeed, in Epidemap, the vector density could be directly informed by field data and used to implement infection risk. This could even be an opportunity to interact with the general public to benefit from local signalling of specific vectors (*Pernat et al., 2021*). Independently of the infection followed, Epidemap can then be used to explore a variety of control scenarios with high resolution at a national level.

## Materials and methods

The approach used in Epidemap couples three models. The first model dispatches each agent to a building, depending on the national demographic distribution (*INSEE, 2019*) and the properties of the buildings (from OSM). The second model determines the buildings that each agent will daily visit and socials interactions with other agents located in the same distant buildings, and is based on that from *Barbosa et al., 2018*. Finally, the third model captures the life-history properties of the infectious disease in infected agents and is based on that from *Sofonea et al., 2021*.

### Geographical structure

We use the freely available OSM database to extract all the points of interest for this study OSM. The accuracy of this database is partially high in France because the official national land registry (the 'Cadastre') database is merged into the OSM database. These databases are formatted in ASCII XML, with a size of approximately 80 GB for France.

OSM labels residential buildings specifically. We use the surface of these buildings and their geographical position to allocate each agent to a 'home' (residency) building. For the mobility model, we also include all the other types of buildings (e.g. hospitals, schools, airports, commercial centres, etc.), where agents can meet.

In our current simulations, the initial database size contains $4.1 \ 10^8$ nodes and $4.8 \ 10^7$ buildings that need to be pre-processed to compute additional characteristics such as building surface, usage, or geographical location, which can be reused in different simulations.

### Demographic model

The Institut National de la Statistique et des Etudes Economiques (INSEE) publishes data with a high-level resolution of the distribution of the French population (*INSEE, 2019*). Here, we use the information about the full population in each city in 2016 to allocate agents to different locations. By combining this database and OSM's, we can compute the number of residents in each building of each city.

More precisely, agents are allocated to buildings proportionally to their floor-surface projection. The number of agents in each building is given by the equation

$$N_k = \left| N_{city} \frac{S_k}{\sum_i S_i} + \alpha \right|. \tag{1}$$

where $N_k$ is the number of agent in the building $k$, $N_{city}$ the number of residents in the city considered, $S_k$ the floor-surface projection of building $k$, $\sum_i S_i$ the total surface of all residential buildings in the city, $|\cdot|$ the entire part of a number, and $\alpha \in [0, 1]$ a scaling parameter such that $\sum_k N_k = N_{city}$.

The age of each agent is randomly generated and follows the age pyramid of the french residents, given by INSEE. In this study, age only affects the probability to develop several symptoms of the infection and, therefore, not the transmission dynamics of the epidemic.

Following the results from the study on individual movement patterns by *Schneider et al., 2013*, we assume that each person can visit two distant buildings per day (where it can meet other agents). Treating the number of visited buildings as a random variable would induce a smaller discretisation time and increase the computing time. Note that, as explained below, in some situations an agent visits less than two buildings per day.

To determine which building is visited by an agent, we first compute its distance $l$ from the home building as the crow flies. Mathematically, $l$ is assumed to follow a log-normal distribution, with $PDF(r) = \textbf{lognorm}(\mu = 2, \sigma = 0.88, r = 0.5l)$. As shown in *Table 1*, this parameterisation yields results that are very consistent with the INSEE data (*INSEE, 2016*).

We then randomly select the building visited by the individual among all the buildings located at a distance $l$ of the agent's home ($\pm 50$ m) using a weighting probability proportional to the floor-surface projection of the buildings. Therefore, larger buildings are visited by a higher amount of people than smaller ones. If no building is found at the randomly generated distance, the agent does not interact with any other agent for this movement round.

Every day, each agent can interact randomly and non-exclusively with other agents present in the same building at the same time. The maximum number of interactions is limited to 17 persons in a distant building and five in a residential building.

**Table 1.** Lognormal randomly choosen values and INSEE statistic of the distance between residential and distant building.
Original datas (**INSEE, 2016**) and fitting.

| | $PDF(0.5l) = \textbf{lognorm}(2, 0.88)$ | **INSEE statistic** |
|---|---|---|
| $lt_{10}$ km | 32.9% | 33.7% |
| 10 km $<X<$20 km | 30.6% | 30.5% |
| 20 km $<X<$30 km | 15.5% | 16.0% |
| 30 km $<X<$50 km | 12.7% | 12.5% |
| 50 km $<X<$100 km | 6.8% | 5.8% |
| $gt_{100}$ km | 1.5% | 1.5% |

As indicated above, following our parsimonious approach, we assume that individual movements do not vary according to age or infection status.

## Infection model

Epidemap is versatile and can be adapted to simulate many infectious disease epidemics. Here, we simulate the case of a respiratory infection, focusing in particular on SARS-CoV-2. Given the life-cycle of the infection, schematised in **Figure 5** we can separate the transmission and clinical dynamics

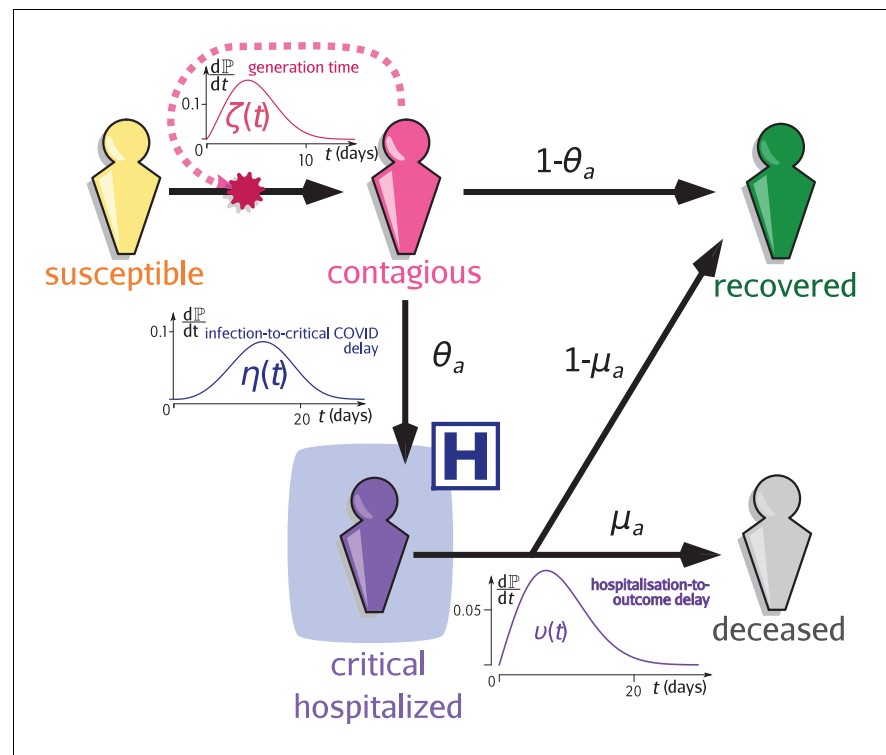

**Figure 5.** Infection model flow chart. Susceptible individuals (yellow figurine), are exposed to viral transmission from contagious individuals (pink figurine). Once infected, a host is more or less contagious depending on the time since contamination according to distribution $\zeta$ called the generation time (and usually parameterised using the empiric serial interval). A fraction $\theta_a$, the value of which depends on the age of the host $a$, will develop a critical infection and be admitted to a hospital (purple figurine) according to the complication delay distribution $\eta$. The complementary fraction $1 - \theta_a$ is assumed to recover with perfect and long-lasting immunity (green figurine). This compartment is also reachable after hospitalisation with the age-dependent probability $1 - \mu_a$, after the discharge delay distribution $\upsilon$. The complementary fraction $\mu_a$ eventually dies from COVID-19. See **Sofonea et al., 2021** for additional details.

because the latter has little to no effect on the former. Indeed, data shows that patients are typically hospitalised 2 weeks after infection. Of the secondary infections, 95% occur within the first 11 days post-infection. Furthermore, only a small fraction of the hosts are hospitalised (less than 1% in France *O'Driscoll et al., 2021*). The parameterisation was done using data from the French COVID-19 epidemic (*Sofonea et al., 2021*; *Tables 2* and *3*).

For each interaction between a susceptible and an infected agent, we assume a constant probability of contamination $b = 5\%$ (*Variant Technical group, 2021*) multiplied by normalised daily infectivity, or generation time, $\zeta(t)$ which follows a Weibull distribution with parameters $k = 2.24$ and $\lambda = 5.42$, with $t$ the number of day since contamination (*Nishiura et al., 2020*). We can simulate the effect of non-pharmaceutical interventions (e.g. mask-wearing) by decreasing this probability. Therefore, the transmission model only requires three parameters, two of which can be informed from contact-tracing data (*Nishiura et al., 2020*).

A fraction $\theta_a$ of infections, where $a$ is the age of the host, become critical (i.e. leading to intensive care unit (ICU) admission and/or death). These individuals have a daily probability $\eta(t)$ to be hospitalised and then a daily probability $\upsilon(t)$ to either recover or die, with probability $1 - \mu_a$ and $1\mu_a$ respectively. All recovered hosts are assumed to be immune to the infection until the end of the simulation.

## Simulation specifications

Because of the strong heterogeneity in the first stages of the epidemic, we perform 100 simulations. To minimise the risk of early extinction of the outbreak, each simulation is seeded by infecting 15 individuals in a building at geographical coordinates 48.87732, 2.32993, which is in the area of Paris (a likely city in France for an introduction given its connectivity to the rest of the world).

The parameters used for these simulations are summarised in *Table 2*. For the type of infections considered, the clinical dynamics do not affect the transmission dynamics and, therefore, the parameters related to the clinical part of the infection are shown in a separate *Table 3*.

The computing code is written in Fortran 90 (F90) with Open Multi-Processing (OMP) approach to parallelise the computation and contains $\simeq 18,000$ lines. A huge effort was made to reduce the memory print of the code, which runs with less than 64 GB Random Access Memory (RAM) for $6.6 \ 10^7$ agents. A full epidemic simulation, which represents approximately 300 days, is performed in 2 hr on a standard personal computer (12-cores AMD Ryzen 9). The statistics are written in ASCII format and the graphical outputs use the compressed Paraview format.

Comparisons with other platforms are indicative because each software has its own properties. For instance, as explained in the introduction, although most softwares do not have the same geographical resolution as Epidemap, some have more detailed social structures such as households. Furthermore, for many softwares, this information is not available. The recent OpenABM platform (*Hinch et al., 2021*) is very transparent about its computation times: takes 1.5 s a 2019 MacBookPro (2.4 GHz Quad-Core Intel Core i5) to simulate 1 day of epidemics for 1 million individuals. To perform our study (100 simulations of 66 million of agents during a year), the computing time would be $\approx 40$ days instead of $\approx 8$ days with Epidemap. Covasim (*Kerr et al., 2021*) would run faster (in approximately 1 day) but does not have spatial structure. In both cases, these estimates assume perfect scaling. Furthermore, the memory required can also raise some problems. For instance, the

**Table 2.** Parameters used for the transmission model in the simulations.
LN stands for log-normal, We for Weibull.

| Name | Description | Value | Reference |
|------|-------------|-------|-----------|
| $N$ | Number of agents (population size) | 6.6E7 | *INSEE, 2019* |
| $H$ | Number of buildings visited during the day (+ home) | 3 | *Schneider et al., 2013* |
| $l$ | Individual dispersal kernel distance | $PDF(0.5 \ l) = LN(2, 0.88)$ | fit of *INSEE, 2016* |
| $N_1$ | Maximum daily number of agent met (distant) | 17 | user-defined |
| $N_2$ | Maximum daily number of agent met (at home) | 5 | user-defined |
| $\zeta(t)$ | Contagiousness $t$ days after infection | $PDF(\zeta) = We(2.24, 5.42)$ | *Nishiura et al., 2020* |
| $b(t)$ | Transmission probability per contact | $5\% \ \zeta(t)$ | *Variant Technical group, 2021* |

**Table 3.** Parameters used for the clinical progression of the infections in the simulations. These parameters do not affect virus spread. LN stands for log-normal, We for Weibull.

| Name | Description | Value | Reference |
|---|---|---|---|
| $\mu$ | Hospital mortality rate | 56% | *Santé Publique France, 2020* |
| $\eta(t)$ | ICU admission daily probability for a severely infected patient | $PDF(\eta) = \text{We}(1.77, 6.52)$ | *Linton et al., 2020* |
| $\upsilon(t)$ | ICU departure daily probability | $PDF(\upsilon) = \text{We}(2, 10)$ | *Linton et al., 2020* |
| $\theta(a)$ | Probability for a severely infected patient of age $a$ to die | $\text{IFR}(a)/\mu$ | *Verity et al., 2020* |

memory required to store the transmission chain data for the contact-tracing component of Open-ABM is $\approx 3$ kB per agent per epidemic week simulated. For one of our runs, that is the simulation of 66 million agents during one year, this would require more than 10 TB of RAM, which is clearly beyond the capabilities of a standard computer. Epidemap requires less than 64 GB for such a run. Covasim would require the same amount of RAM but for shorter simulations (100 days) and no spatial structure. *Aleta et al., 2020* and *Rockett et al., 2020* do not seem to provide indications regarding the computing speed of their framework.

## Acknowledgements

The authors thank the Région Occitanie and the ANR for funding (PhyEpi grant).

## Additional information

### Funding

| Funder | Grant reference number | Author |
|---|---|---|
| Région Occitanie Pyrénées-Méditerranée< | | Samuel Alizon |
| Agence Nationale de la Recherche | PhyEpi ANR project | Samuel Alizon |

The funders had no role in study design, data collection and interpretation, or the decision to submit the work for publication.

### Author contributions

Olivier Thomine, Data curation, Software, Formal analysis, Investigation, Visualization, Methodology; Samuel Alizon, Resources, Formal analysis, Validation, Methodology, Writing - original draft, Writing - review and editing; Corentin Boennec, Formal analysis, Validation, Visualization, Methodology, Writing - original draft, Writing - review and editing; Marc Barthelemy, Data curation, Formal analysis, Methodology, Writing - original draft, Writing - review and editing; Mircea Sofonea, Resources, Formal analysis, Validation, Visualization, Methodology, Writing - original draft, Writing - review and editing

### Author ORCIDs

Olivier Thomine (iD) https://orcid.org/0000-0002-4847-3224

### Decision letter and Author response

Decision letter https://doi.org/10.7554/eLife.71417.sa1
Author response https://doi.org/10.7554/eLife.71417.sa2

## Additional files

### Supplementary files
• Transparent reporting form

## Data availability

The raw data associated with the 100 simulations performed and the R scripts used to generate the figures are available from the Zenodo repository at https://zenodo.org/record/5542171 (results.zip).

The following dataset was generated:

| Author(s) | Year | Dataset title | Dataset URL | Database and Identifier |
|---|---|---|---|---|
| Thomine O, Alizon S, Barthelemy M, Boennec C, Sofonea MT | 2021 | Emerging dynamics from high-resolution spatial numerical epidemics | https://doi.org/10.5281/zenodo.5542171 | Zenodo, 10.5281/zenodo.5542171 |

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
