## [Decision Letter]

**Acceptance summary:**

This work presents a dynamic infectious disease transmission model using geospatial data to structure transmission, using SARS-CoV-2 transmission in France as an example. The model allows for the incorporation of fine grain spatial heterogeneity and a large number of simulated individuals, providing a computationally efficient alternative to traditional agent-based models and a more realistic geographical mixing structure than traditional compartmental model. The Epidemap framework has many potential uses for supporting infectious disease planning and response activities beyond the SARS-CoV-2. The work will be of interest to infectious disease modelers, epidemiologists, and public health decision-makers working in epidemic outbreak management.

**Decision letter after peer review:**

Thank you for submitting your article "Emerging dynamics from high-resolution spatial numerical epidemics" for consideration by *eLife*. Your article has been reviewed by 3 peer reviewers, one of whom is member of our Board of Reviewing Editors, and the evaluation has been overseen by a Senior Editor. The reviewers have opted to remain anonymous.

Essential revisions:

1. The paper does not currently meet *eLife*'s policy regarding Availability of Data, Software, and Research Materials. The revision should include material to meet this policy.

2. The paper should be reformatted using a more traditional structure to improve readability (Introduction, Methods, Results, Discussion).

3. Include further explanations and discussion regarding how age is integrated into the model, or how the model could be extended to include host heterogeneity by age.

4. Include further description and explanation of how the distance kernel and mobility kernel were modeled.

5. There is currently very little discussion on how the results of this study compare with results from previous traditional models. There should be more discussion on how this study fits into previous literature on this topic, including the citation of key previous papers that have examined issues of spatial heterogeneity.

6. Include more high-level details regarding the model structure from the appendix in the main text to help the reader understand the structure without having to refer to previously published papers. The reviewers provide some suggestions in their reviews for which elements could be included.

7. The paper currently includes specifications regarding the computational resources needed for this platform, but should also include further discussion on computational resources required by traditional compartmental and agent-based models to so that the reader can appreciate the difference. Further discussion on how model results and computational resources compare with traditional compartmental and agent-based models.

8. Further discussion around limitations of the tool, particularly in the case of application to other infections where distance alone may not sufficiently capture transmission patterns.

*Reviewer #1:*

In this work, the authors present an infectious disease transmission model using geospatial data to structure transmission. Their aim was to produce a stochastic agent-based model that integrates geographic structure and infection natural history with sufficient realism without being too computationally demanding. By integrating map and daily mobility information, the work shows that interesting infection dynamics may occur at different geographical levels when geographic structure is considered in the context of transmission. Models incorporating geographical information are likely to be increasingly valuable in the future, as the COVID-19 pandemic has highlighted the need to consider regional and local needs when implementing public health measures against a pandemic pathogen.

Some of the model strengths include the use of detailed geographical information, and a minimal number of parameters needed to inform the model. A wide range of natural history of infection models can potentially be integrated to represent different agents with different transmission and immunity profiles. Perhaps one of the weaknesses of the model is the lack of age stratification, which would increase the realism of the model and provide important epidemiological information from a public health perspective. Age has turned out to be an important variable in tracking the COVID-19 pandemic impact and public health response. While the authors mention the model tracks the age of participants, the parameters and behaviors of agents do not appear to depend on age. A further discussion of how this model could be extended to include more heterogeneity in the movement patterns of agents would be useful, and whether the inclusion of further complexity would substantially increase the computational burden of simulations. There is also no data presented regarding the hospitalization component of the model, which is briefly described but not explored.

I think one of the major contributions of this work is the illustration of how high-resolution geographical data can be integrated into infectious disease models. These methods are likely to be of high interest to other infectious disease modelers, and to public health experts working in epidemic outbreak management.

– The paper does not currently meet *eLife*'s policy regarding Availability of Data, Software, and Research Materials (https://submit.elifesciences.org/html/*eLife*_author_instructions.html#policies). I can appreciate that the dataset generated by the model is too large to be made available in free data repositories. However, this does not preclude increasing the reproducibility and availability of the data. In cases where the data can't be made available, it is up to the authors to explain in the manuscript the restrictions on the dataset or materials and why it is not possible to give public access. They must also provide a description of the steps others can follow to request access to the data or materials if they are interested. It is also good practice to provide access to data and materials for which the constraints do not apply. For example, what I have seen in similar cases with large or un-shareable datasets is that the authors would provide the necessary code to reproduce figures and tables in the manuscript with a smaller simulated dataset. Often also the dataset can be broken down into smaller datasets of the processed data necessary to reproduce each figure. While Figure 3 might be problematic due to the large number of observations, Figures 2 and 4 would likely be amenable to this as each panel appears to only display the results from a couple hundred data points. I suggest the authors consider this option. This should all be included in the data availability statement as well.

– Please further discuss how the model could be built on to add further demographic stratifications such as age to natural history/daily mobility patterns and interactions between agents. Would the addition of further stratifications severely affect the computational burden?

– The authors mention a parameter regarding hospitalization probability and severity of infection; however, these parameters are not included in the table of parameters in the appendix. It would seem to me there are more than 6 parameters in the model then. It is unclear why these parameters were added, as they are not explored in the results or mentioned very much in the text. Some more discussion or results regarding this component of the model would be warranted.

– There is little discussion of other models which have implemented geographical structure, and how this model compares with those. I am not very familiar with this literature, but I find it hard to believe that none have tried to implement some geographical component. Some more discussion on how this approach is innovative or different compared to what has been done in the past would be useful.

– It would be useful to include the names of the scientific papers cited in the reference list, most of these are not full references.

*Reviewer #2:*

This work provides a new general tool for studying the chains and patterns of transmission of infectious diseases. It addresses the limitations of mathematical models and the Agent-Based Simulation platforms in public health by using High-Performance Computing techniques, high-resolution spatial data, and complex mobile models. Based on the results of 100 stochastic simulations, the basic reproduction number, the duration of the disease and the final total epidemic size were obtained at the national level, which shows the importance of the geographical structure. Meanwhile, the influence of the distance from origin and the density of the region on the epidemic was shown at the district level. Finally, this paper also shows that the importance of super-spreading events varies according to the stage of the epidemic.

1. In the Introduction part, the authors mentioned some work on COVID-19 based on mathematical modelling. However, some existed work are not well respected. Please see some papers: Short-term predictions and prevention strategies for COVID-19: A model-based study, Applied Mathematics and Computation, 2021; Analysis of COVID-19 transmission in Shanxi Province with discrete time imported cases, MBE, 2020; An investigation of transmission control measures during the first 50 days of the COVID-19 epidemic in China, Science, 2020; Transmission dynamics of COVID-19 in Wuhan, China: effects of lockdown and medical resources, Nonlinear Dynamics, 2020.

2. For readability, please give a brief description and introduction of a distance kernel and a mobility kernel mentioned in line 52 in the text.

3. In line 55, the author mentioned that the simulation tracked the age of the individual, but this was not further described and shown in the text, as well as the description of the relevant simulation result. Given the importance of age to COVID-19, further research on age should be conducted.

4. This paper demonstrates the power and flexibility of the Epidemap platform through the application of COVID-19 in France. However, all the results obtained in the paper are obtained through numerical simulation, the authors should compare them with the real data at the national and district level of France to further prove the rationality, practical application and authenticity of this method.

*Reviewer #3:*

Thomine et al., have a developed a new tool for modeling infectious diseases which can consider fine grain geographic movements of tens of millions of individual agents (simulated persons), thereby enabling more realistic simulations (compared to SIR models) without the excessive computational demands required by traditional agent-based modeling approaches. Impressively, this tool, Epidemap, was able to simulate one year of daily interactions and epidemic growth trajectories for the entire population France (approximately 65 million people) in less than two hours using a standard high performance computer.

The authors present the example of an uncontrolled SARS-CoV-2 epidemic in France and identified spatio-temporal differences in disease and transmission dynamics that would not be discernable using naïve SIR modeling approaches and would be extremely computationally demand to complete using traditional ABM methods. These observations included a distinct bimodal pattern, in which each peak was comprised of different localities; a strong correlation between the timing of the epidemic peak in different regions and its distance from the point of epidemic origin; important differences in disease dynamics based on population density; and unique insights regarding secondary attack rates measured at the individual level (i.e., the reproductive number). These observations could support evidence-informed targeting of public health measures to optimize the impact of mitigation measures and support health care planning. This tool could also have great applicability to the study of other respiratory infections, particularly if additional features further enhancing the realism of the simulations, such as assigning children to schools, can be added without substantially increasing computational demands. The visual component of this tool is an especially nice feature, which could greatly support knowledge translation activities with decision-makers and planners.

The rationale for the development of this tool is clear (and important), the conclusions of this manuscript are supported by the data, and the paper is well-written. The included figures, particularly Figure 1, are very easy to follow and nicely display the key take-away messages.

The methods section could benefit from additional details to better able the reader to understand the development of this tool, specifically:

1) Please provide adequate details regarding the fundamentals of this approach in the main manuscript text. The material provided in S1 Supplementary Methods is critical to understanding this tool, particularly the summary statement regarding the three specific models. For example, the manuscript refers to the epidemiological model, but the reader must refer to a reference to learn more. Providing some high-level details regrading the model and the hospital data (including how they were used to parametrize the model) would be helpful. Similarly, it is stated that the disease progression model follows that of reference 10 – having a figure included in the manuscript would be helpful – and the daily reproduction numbers were based on a method from 18 – a brief description would be appreciated.

2) How do these findings compare to traditional SIR or ABM models? Understandably, it may be too computationally demanding to run a traditional ABM for the entire population of France and would likely be out of scope for this study. For context, it would be useful to provide an estimate of the time and computational resources demanded by traditional approaches. If running these other models are possible, a comparison of the insights provided across these 3 methods would be highly valuable – particularly if there are large differences.

3) The probability of encounters is based only on distance. As the authors state, this assumption may not hold for other countries (e.g. the US where air travel is more important). This assumption may also not hold for other infections. For example, the transmission dynamics of pediatric respiratory viruses are more influenced by neighbourhood-level patterns of movement – whereas diseases of adults are more heavily influenced by larger-scale geographic patterns. Please provide the reader with more context around this limitation.

– Abstract: I'm not sure if "computational-efficient" is grammatically correct – suggested revision: computationally efficient.

– The introduction section could be strengthened by first introducing the idea of a mathematical model – and their uses – before discussing their limitations. Could you provide the reader with an explicit example of where these models have failed because they did not contain the features of an ABM (or Epidemap)? There are several examples from the COVID-19 pandemic and Ebola epidemics that readers would be familiar with and would allow them to immediately appreciate the importance of the current work.

– Introduction: You state that SIR models ignore spatial contact patterns. Though the naïve SIR model does, most SIR models are age-structured and include some sort of contact matrix (e.g. POLYMOD). Suggest rewording to "and oversimplifies contact patterns".

– Regarding the following statement: "A third limit resides in the way geographic structure is implemented into the simulation (but see (7))." Please clarify what is meant without the reader referring to a reference.

– Such a model is likely only relevant to the study of respiratory viruses, this should be stated as a limitation – or, if modifications can be made to enable the study of other infectious diseases (e.g. STIs), this should be highlighted as a strength of Epidemap.

– The readability of the manuscript would also benefit from a more traditional structure, i.e., sub-headers in the abstract and main text for background, methods, results, and discussion. Similarly, the funding statement is provided as reference. This, along with a conflict of interest statement, should be explicitly provided in-text.

– In the supplement, you refer to the spread of COVID-19. Recommended revision: SARS-CoV-2.

– To enhance the clarity of Figure 2, it would be helpful to line-up the x-axes of (a) and (b).

Specific aspects of the methodology that are not clear from the manuscript or supplemental:

– The justification for some modeling choices has not been provided and it is not clear what impact, if any, this would have had on the results. Namely, what was the rationale for initializing the model with 15 infected individuals in Paris and aligning the axis for Figure 2 based a value of 700 ICU beds. Assumedly, the choice for Paris is due to this being the most likely place for importation, but this is not clear. The choices for the other two values appears arbitrary.

– It is not clear how the interaction model accounts for household and school/workplace encounters. For example, are these included in the random movement or separately? Does the risk of transmission differ in these contexts? These dynamics would be quite different than a random encounter at, for example, the grocery store. Similarly, can transmission occur within hospitals?

– The age of contacts is recorded, but it is not clear how/if this information is incorporated into the simulation; e.g. differences in disease severity profiles on the basis of age.

– How were the point estimates and 95% CI calculated?

---

## [Author Response]

Essential revisions:1. The paper does not currently meet eLife's policy regarding Availability of Data, Software, and Research Materials. The revision should include material to meet this policy.

We posted on the Zenodo server the raw outputs of the simulation data (ASCII format), as well as the R scripts used to generate the figures.

The availability of the data and scripts is now mentioned in the manuscript.

2. The paper should be reformatted using a more traditional structure to improve readability (Introduction, Methods, Results, Discussion).

We now highlight the article structure and included a clear Methods section.

3. Include further explanations and discussion regarding how age is integrated into the model, or how the model could be extended to include host heterogeneity by age.

This is an individual-based simulation so each individual has an age, which is randomly chosen based on the French demographics data from INSEE (the French statistics institute). In the simulation, the age only affects the clinical part of the model, i.e. the probability to develop severe symptoms and be hospitalised. There are several ways in which age could be included in the model. One possibility could be to implement realistic household structures using existing data. Furthermore, regarding the daily mobility patterns, we could simulate school attendance by imposing that during the week days all children visit the nearest school (a piece of information that is available through the OpenStreetMaps/OSM data). Similarly, for older individuals, we could simulate residency in age care facilities, although this data may require more effort to be extracted from OSM. These examples were not implemented in these first simulations to avoid increasing the number of free parameters but EPIDEMAP does offers many possibilities to investigate the role of age in epidemic spread and simulate control scenarios.

We now mention some of the scenarios that could be explored in EPIDEMAP using the age structure already implemented.

4. Include further description and explanation of how the distance kernel and mobility kernel were modeled.

We apologize for putting all the details about the distance kernel in the Appendix. As we now detail in the Methods section, all individuals are assigned to an OSM building (their ‘home’). Each day, they can visit 2 additional buildings and this is where the distance kernel matters. These buildings are chosen at random on a circle the radius of which is drawn at random in a lognormal distribution. If more than one building intersects with the circle, we draw a building at random. If there is no building on the circle, this movement is cancelled.

We now discuss the distance kernel in the new Methods section.

5. There is currently very little discussion on how the results of this study compare with results from previous traditional models. There should be more discussion on how this study fits into previous literature on this topic, including the citation of key previous papers that have examined issues of spatial heterogeneity.

This is a delicate issue because there is a wealth of individual based models (IBM) but very few nationwide models are performed with a realistic geographical mapping. Indeed, spatially explicit individual-based model tend to have a higher level of granularity (e.g. with cells or districts in which individuals interact but where spatial structure does not matter). There are some exception, such as https://link.springer.com/article/10.1186/1471-2334-11-115

We now discuss in more details studies that investigated spatial heterogeneity and how our results on spatial epidemic spread relate to these.

6. Include more high-level details regarding the model structure from the appendix in the main text to help the reader understand the structure without having to refer to previously published papers. The reviewers provide some suggestions in their reviews for which elements could be included.

All the technical descriptions of the model were in the Appendix, which we agree was not ideal for readers who want to know more about the specificity of our simulator.

We added a Methods section to the main text of the manuscript.

7. The paper currently includes specifications regarding the computational resources needed for this platform, but should also include further discussion on computational resources required by traditional compartmental and agent-based models to so that the reader can appreciate the difference. Further discussion on how model results and computational resources compare with traditional compartmental and agent-based models.

In short, we are not aware of any existing individual based model that could handle so many individuals (66 million) in a spatially-explicit context on a similar (regular) desktop. For example, according to its specifications, the recent platform openABM takes 1.5s on a 2019 MacBookPro (2.4GHz QuadCore Intel Core i5) to simulate 1 days of epidemics for 1 million individuals. https://doi.org/10. 1371/journal.pcbi.1009146 It would therefore take 10 hours to simulate one of our runs (and without spatially explicit setting), assuming a perfect-scaling of their approach, where Epidemap took less than 2 hours. Furthermore, their approach requires «3*kB* per agent for tracing storage purposes for a 7 days pandemic. One simulation equivalent to ours would therefore require more than 10TB of RAM with this platform, which is clearly beyond the capabilities of a standard computer, where Epidemap needs less than 64 GB. Importantly, openABM has features that are not yet implemented in Epidemap, such as contact tracing and these may require additional memory. Furterhmore, their authors published these specifications, which is not the case of many simulators.

At the end of the Methods section, now compare more explicitly our computing power to more traditional agent-based models, while also pointing out that other models have some properties that are not (yet) included in Epidemap.

8. Further discussion around limitations of the tool, particularly in the case of application to other infections where distance alone may not sufficiently capture transmission patterns.

We indeed assumed that transmission occurs upon daily contacts, which better corresponds to respiratory infections. Simulating the spread of infections with other transmission routes could indeed be challenging. For instance, modelling sexual transmission would require to simulate partnerships in a more detailed way (see e.g. https://royalsocietypublishing.org/doi/full/10.1098/ rsif.2011.0131). Similarly, epidemics from vector-borne diseases would likely require an additional layer to simulate the vector demographics. However, would represent interesting future extensions for Epidemap, especially the vector-borne transmission mode.

We now better stress that the current implementation of the model is better suited to respiratory infections and discuss potential extensions to infections with different transmission routes.

Reviewer #1:In this work, the authors present an infectious disease transmission model using geospatial data to structure transmission. Their aim was to produce a stochastic agent-based model that integrates geographic structure and infection natural history with sufficient realism without being too computationally demanding. By integrating map and daily mobility information, the work shows that interesting infection dynamics may occur at different geographical levels when geographic structure is considered in the context of transmission. Models incorporating geographical information are likely to be increasingly valuable in the future, as the COVID-19 pandemic has highlighted the need to consider regional and local needs when implementing public health measures against a pandemic pathogen.Some of the model strengths include the use of detailed geographical information, and a minimal number of parameters needed to inform the model. A wide range of natural history of infection models can potentially be integrated to represent different agents with different transmission and immunity profiles. Perhaps one of the weaknesses of the model is the lack of age stratification, which would increase the realism of the model and provide important epidemiological information from a public health perspective. Age has turned out to be an important variable in tracking the COVID-19 pandemic impact and public health response. While the authors mention the model tracks the age of participants, the parameters and behaviors of agents do not appear to depend on age. A further discussion of how this model could be extended to include more heterogeneity in the movement patterns of agents would be useful, and whether the inclusion of further complexity would substantially increase the computational burden of simulations. There is also no data presented regarding the hospitalization component of the model, which is briefly described but not explored.I think one of the major contributions of this work is the illustration of how high-resolution geographical data can be integrated into infectious disease models. These methods are likely to be of high interest to other infectious disease modelers, and to public health experts working in epidemic outbreak management.

Thank you for the positive assessment of our work!

Regarding the age structure, we apologise for the lack of clarity (which is probably due to the fact that most of the model details were in the Appendix). Indeed, the age stratification was already implemented (an individual is currently defined by her/his home building and her/his age). The age also affects the probability to develop a severe infection in the model (which is not shown in the model for focusing reasons). What we did not yet implement is a differential behaviour based on age, e.g. the fact that children attend school, or the fact that older people visit less buildings per day that younger people. As indicated to the Editor, these variations can readily be added but they also require solid data to avoid an inflation in the number of free parameters in the model.

We clarified the current role of individual age in the model.

We now discuss perspectives regarding heterogeneity in individual behaviour, especially depending on their age.

– The paper does not currently meet eLife's policy regarding Availability of Data, Software, and Research Materials (https://submit.elifesciences.org/html/eLife_author_instructions.html#policies). I can appreciate that the dataset generated by the model is too large to be made available in free data repositories. However, this does not preclude increasing the reproducibility and availability of the data. In cases where the data can't be made available, it is up to the authors to explain in the manuscript the restrictions on the dataset or materials and why it is not possible to give public access. They must also provide a description of the steps others can follow to request access to the data or materials if they are interested. It is also good practice to provide access to data and materials for which the constraints do not apply. For example, what I have seen in similar cases with large or un-shareable datasets is that the authors would provide the necessary code to reproduce figures and tables in the manuscript with a smaller simulated dataset. Often also the dataset can be broken down into smaller datasets of the processed data necessary to reproduce each figure. While Figure 3 might be problematic due to the large number of observations, Figures 2 and 4 would likely be amenable to this as each panel appears to only display the results from a couple hundred data points. I suggest the authors consider this option. This should all be included in the data availability statement as well.

As indicated in our response to the Editor, we posted the raw data on Zenodo, as well as the data and scripts used to generate the figures.

– Please further discuss how the model could be built on to add further demographic stratifications such as age to natural history/daily mobility patterns and interactions between agents. Would the addition of further stratifications severely affect the computational burden?

We do not expect additional stratification to affect computational burden if the traits involved are already implemented. The main risk for this study is the paramterisation issue. For instance, as we show for superspreading events, even without factoring these in explicitly in the model, they partly emerge from the simulations. Therefore, forcing a stratification or a heterogeneity without rich data may lead to an over-parameterisation of the model. As indicated in our response to the Editor, one of the most interesting extensions, which could be supported with data, is the household structure. This would naturally lead to additional age-stratification. Differences in behaviour could then be added on top of this.

We now discuss the main difficulty to add details to the model (which has more to do with the risk of over-parameterising the model) and which extensions are the most promising.

– The authors mention a parameter regarding hospitalization probability and severity of infection; however, these parameters are not included in the table of parameters in the appendix. It would seem to me there are more than 6 parameters in the model then. It is unclear why these parameters were added, as they are not explored in the results or mentioned very much in the text. Some more discussion or results regarding this component of the model would be warranted.

The model was developed in the context of the COVID-19 epidemic in France, and it does include a clinical component. The reviewer is entirely correct that the hospital (and mortality) component require additional parameters. However, these do not affect the results shown since, in the case of COVID-19, the hospital side of the epidemic has little effect on the spread of the infection in the general population (a small fraction is hospitalised and severe cases are admitted into hospitals 14 days after infection on average, whereas after 11 days, more than 95% of the secondary infections have already occurred). Overall, the three parameters related to the infection model that are required for the transmission model are the probability of transmission per contact and the two parameters capturing the serial interval distribution.

We now specify that the hospital and mortality extension of the model require additional parameters and add a second table showing the parameters related to this more clinical part of the model. We also added a video showing the number of hospital admission in France over the course of the simulation.

– There is little discussion of other models which have implemented geographical structure, and how this model compares with those. I am not very familiar with this literature, but I find it hard to believe that none have tried to implement some geographical component. Some more discussion on how this approach is innovative or different compared to what has been done in the past would be useful.

As indicated above, relatively few individual based model include detailed geographical structure (especially at a national level). In most models, this structure is discretised into subunits and a metapopulation approach is applied where individuals interact within their unit/population and migrate between populations. Some frameworks allow to perform similar simulations with high spatial resolution, e.g. the GAMA platform, but they tend to cover smaller geographical areas (partly to limit the number of individuals).

We now discuss more extensively the literature on explicit geographical structure in individual based models in the Introduction with a focus on recent COVID-19 models.

– It would be useful to include the names of the scientific papers cited in the reference list, most of these are not full references.

The initial reference style used was indeed poorly informative. We now switched to eLife’s LATEX template.

Reviewer #2:This work provides a new general tool for studying the chains and patterns of transmission of infectious diseases. It addresses the limitations of mathematical models and the Agent-Based Simulation platforms in public health by using High-Performance Computing techniques, high-resolution spatial data, and complex mobile models. Based on the results of 100 stochastic simulations, the basic reproduction number, the duration of the disease and the final total epidemic size were obtained at the national level, which shows the importance of the geographical structure. Meanwhile, the influence of the distance from origin and the density of the region on the epidemic was shown at the district level. Finally, this paper also shows that the importance of super-spreading events varies according to the stage of the epidemic.

Thank you for this accurate summary.

1. In the Introduction part, the authors mentioned some work on COVID-19 based on mathematical modelling. However, some existed work are not well respected. Please see some papers: Short-term predictions and prevention strategies for COVID-19: A model-based study, Applied Mathematics and Computation, 2021; Analysis of COVID-19 transmission in Shanxi Province with discrete time imported cases, MBE, 2020; An investigation of transmission control measures during the first 50 days of the COVID-19 epidemic in China, Science, 2020; Transmission dynamics of COVID-19 in Wuhan, China: effects of lockdown and medical resources, Nonlinear Dynamics, 2020.

Searching the keywords "mathematical + modelling + COVID-19" in the Web of Science led to 2,765 references and reviewing all of these is clearly beyond the scope of this study. Furthermore, we are unsure as to why the reviewer singled out these studies because none of these correspond to individual based models.

1. Nadim et al., (2021, Applied Mathematics and Computation) develop a very basic ODE model which is very similar to dozens of models published before,

2. Li et al., (2020, Mathematical Biosciences and Engineering) use a simpler discrete-time model to analyse the cumulative number of cases in China, a strong limitation being that cumulative number of cases is easier to fit than incidence time series,

3. Tian et al., (2020, Science) analyse some valuable data but from a modelling perspective, all they seem to be implementing is a mean field SEIR model,

4. Sun et al. (2020, Nonlinear Dynamics) is very similar to the model by Nadim et al. (a basic ODE model with 5 compartments) and, as Li et al., is used to fit cumulative incidences.

We clarified the manuscript to underline that the originality of this study is not about COVID-19 epidemiological modelling and that there are few individual based studies that feature explicit geographical location.

2. For readability, please give a brief description and introduction of a distance kernel and a mobility kernel mentioned in line 52 in the text.

As indicated above, we apologise for the lack of clarity and the fact that most of the Methods we in Appendix.

We added a Methods section to describe the technical details of the simulations.

3. In line 55, the author mentioned that the simulation tracked the age of the individual, but this was not further described and shown in the text, as well as the description of the relevant simulation result. Given the importance of age to COVID-19, further research on age should be conducted.

This point was also raised by the other reviewer. Age is indeed an important factor regarding the infection fatality ratio of many respiratory infections. It was already included in the simulations but we indeed chose not to underline the results regarding mortality and hospital admissions to focus on the simulator itself.

We clarified the role of age and now show the hospital admission data and the mortality data in the Appendix.

4. This paper demonstrates the power and flexibility of the Epidemap platform through the application of COVID-19 in France. However, all the results obtained in the paper are obtained through numerical simulation, the authors should compare them with the real data at the national and district level of France to further prove the rationality, practical application and authenticity of this method.

The goal of the simulator is not to recreate the true epidemic, which would require to factor in all the public health and political decisions. Furthermore, the intensity of the computations and the level of detail make it also impractical to use such a simulator for parameter estimation (simpler compartmental models are more suited to analyse past data). The goal of a detailed simulator as EPIDEMAP is to analyse different scenarios using the most parsimonious approach (besides the hospital side of the epidemics, our model only requires 6 parameters) to simulate epidemics. Furthermore, our analyses on the resulting patterns indicate high level of biological realism, as illustrated for instance by the distribution of ….

We clarified at the end of the Introduction that the goal of EPIDEMAP is not to simulate the epidemic as it occurred or to perform statistical parameter estimation. We explain that it is a tool to explore epidemic scenarios with a high level of resolution.

Reviewer #3:Thomine et al., have a developed a new tool for modeling infectious diseases which can consider fine grain geographic movements of tens of millions of individual agents (simulated persons), thereby enabling more realistic simulations (compared to SIR models) without the excessive computational demands required by traditional agent-based modeling approaches. Impressively, this tool, Epidemap, was able to simulate one year of daily interactions and epidemic growth trajectories for the entire population France (approximately 65 million people) in less than two hours using a standard high performance computer.The authors present the example of an uncontrolled SARS-CoV-2 epidemic in France and identified spatio-temporal differences in disease and transmission dynamics that would not be discernable using naïve SIR modeling approaches and would be extremely computationally demand to complete using traditional ABM methods. These observations included a distinct bimodal pattern, in which each peak was comprised of different localities; a strong correlation between the timing of the epidemic peak in different regions and its distance from the point of epidemic origin; important differences in disease dynamics based on population density; and unique insights regarding secondary attack rates measured at the individual level (i.e., the reproductive number). These observations could support evidence-informed targeting of public health measures to optimize the impact of mitigation measures and support health care planning. This tool could also have great applicability to the study of other respiratory infections, particularly if additional features further enhancing the realism of the simulations, such as assigning children to schools, can be added without substantially increasing computational demands. The visual component of this tool is an especially nice feature, which could greatly support knowledge translation activities with decision-makers and planners.The rationale for the development of this tool is clear (and important), the conclusions of this manuscript are supported by the data, and the paper is well-written. The included figures, particularly Figure 1, are very easy to follow and nicely display the key take-away messages.

Many thanks for this very detailed and enthusiastic presentation of our work!

The methods section could benefit from additional details to better able the reader to understand the development of this tool, specifically:1) Please provide adequate details regarding the fundamentals of this approach in the main manuscript text. The material provided in S1 Supplementary Methods is critical to understanding this tool, particularly the summary statement regarding the three specific models. For example, the manuscript refers to the epidemiological model, but the reader must refer to a reference to learn more. Providing some high-level details regrading the model and the hospital data (including how they were used to parametrize the model) would be helpful. Similarly, it is stated that the disease progression model follows that of reference 10 – having a figure included in the manuscript would be helpful – and the daily reproduction numbers were based on a method from 18 – a brief description would be appreciated.

As indicated in our replies to the other reviewers, we heartily acknowledge that the detailed methods were difficult to follow being mostly in the Appendix. In the new methods section we added to the main text, we further describe the model used for disease progression. Furthermore, we also show the corresponding outputs for hospital admission data and mortality data in the Appendix.

We added a new Methods section, in which we describe the disease progression model and the tools used to estimate the reproduction number.

2) How do these findings compare to traditional SIR or ABM models? Understandably, it may be too computationally demanding to run a traditional ABM for the entire population of France and would likely be out of scope for this study. For context, it would be useful to provide an estimate of the time and computational resources demanded by traditional approaches. If running these other models are possible, a comparison of the insights provided across these 3 methods would be highly valuable – particularly if there are large differences.

If the reviewers’ suggestion is to compare our model to a classical ABM with the whole French population without spatial structure, the answer is each because this should yield the same results as model based on the equations used to simulate disease progression and transmission. Such a framework is already implemented in the COVIDSIM model (Sofonea et al., 2021 *Epidemics*). The only thing to do is to have the same basic reproduction numbers (*R*_0_) in EPIDEMAP and in this model. In the Appendix, we now show the dynamics of the daily case incidence for the transmission model without spatial structure and for the EPIDEMAP simulations.

We added a figure in the Appendix (Supplementary Figure S1) to show the epidemiological dynamics using the same parameters but in a setting without spatial structure.

3) The probability of encounters is based only on distance. As the authors state, this assumption may not hold for other countries (e.g. the US where air travel is more important). This assumption may also not hold for other infections. For example, the transmission dynamics of pediatric respiratory viruses are more influenced by neighbourhood-level patterns of movement – whereas diseases of adults are more heavily influenced by larger-scale geographic patterns. Please provide the reader with more context around this limitation.

We completely agree with this limitation. In the current simulations, we assume that host age does not affect susceptibility to the infection or contagiousness. Adding these effects would be very valuable to focus on specific infections. It would, however, require to improve some features of the model, especially the household structure and potentially also the mobility patterns (by having children attending schools). Such an implementation could be particularly interesting because it might uncover interactions between age-based differences in spatial/mobility patterns and in biology (sensitivity to infection and contagiousness).

When discussing potential extensions of the model to include household structure and age-based mobility patterns, we now explain how this could be particularly useful to better understand infections that spread differently across age groups.

– Abstract: I'm not sure if "computational-efficient" is grammatically correct – suggested revision: computationally efficient.

We made the changes.

– The introduction section could be strengthened by first introducing the idea of a mathematical model – and their uses – before discussing their limitations. Could you provide the reader with an explicit example of where these models have failed because they did not contain the features of an ABM (or Epidemap)? There are several examples from the COVID-19 pandemic and Ebola epidemics that readers would be familiar with and would allow them to immediately appreciate the importance of the current work.

Pointing out the usefulness of models is a good point. Epidemap’s key feature is without any doubt the detailed geographical structure. One of the patterns it could help to explain is the great heterogeneity in total epidemic size across the country. As we show here, human density strongly impacts this spread.

We highlight the importance of mathematical models in the introduction and mention specific examples in the discussion that show the importance of ABM with high geographical resolution.

– Introduction: You state that SIR models ignore spatial contact patterns. Though the naïve SIR model does, most SIR models are age-structured and include some sort of contact matrix (e.g. POLYMOD). Suggest rewording to "and oversimplifies contact patterns".

We agree and changed the formation.

– Regarding the following statement: "A third limit resides in the way geographic structure is implemented into the simulation (but see (7))." Please clarify what is meant without the reader referring to a reference.

What we meant was that in many simulations the spatial structure does not correspond to the actual geographical structure and is, for instance, simplified into a network or a lattice structure.

We reformulated the sentence for improved clarity.

– Such a model is likely only relevant to the study of respiratory viruses, this should be stated as a limitation – or, if modifications can be made to enable the study of other infectious diseases (e.g. STIs), this should be highlighted as a strength of Epidemap.

This point was also raised by the Editor. While in theory infections with other types of transmission modes could be studied, these would require slight modifications in the main code. However, there is a lot of potential, for instance to simulate vector-borne diseases by factoring in the environmental distribution of the vector.

We mention how the current implementation of EPIDEMAP is adapted to respiratory infections and discuss how other types of infections could be modelled.

– The readability of the manuscript would also benefit from a more traditional structure, i.e., sub-headers in the abstract and main text for background, methods, results, and discussion. Similarly, the funding statement is provided as reference. This, along with a conflict of interest statement, should be explicitly provided in-text.

We now use the more classical structure from *eLife*.

– In the supplement, you refer to the spread of COVID-19. Recommended revision: SARS-CoV-2.

We agree and made the change.

– To enhance the clarity of Figure 2, it would be helpful to line-up the x-axes of (a) and (b).

The motivation for having different axes was to zoom on the two waves in order to better see the specific contribution of each region which are otherwise too thin.

The x-axes of the two panels are now aligned.

Specific aspects of the methodology that are not clear from the manuscript or supplemental:– The justification for some modeling choices has not been provided and it is not clear what impact, if any, this would have had on the results. Namely, what was the rationale for initializing the model with 15 infected individuals in Paris and aligning the axis for Figure 2 based a value of 700 ICU beds. Assumedly, the choice for Paris is due to this being the most likely place for importation, but this is not clear. The choices for the other two values appears arbitrary.

The initialization was done with 15 infected individuals in a small area to minimise the risk of early extinction. Paris was indeed chosen for the increased risk of importation. We indeed used a threshold in ICU bed occupancy to align the dynamics. This was made because the initial stages of an epidemic are very stochastic. By aligning the dynamics, we can better visualise the common features between all these simulations (in each of which 60 million agents live a different daily life for a year). The threshold value itself (700 ICU beds) originates from the COVID-19 epidemics: it was the value when the first lockdown was implemented on March 17, 2020. In a way, we align each simulation with the last event before the first lockdown.

We now explain in the methods the reason for choosing specific values for initialising the simulations and aligning the dynamics. We also better explain the motivation for aligning these dynamics.

– It is not clear how the interaction model accounts for household and school/workplace encounters. For example, are these included in the random movement or separately? Does the risk of transmission differ in these contexts? These dynamics would be quite different than a random encounter at, for example, the grocery store. Similarly, can transmission occur within hospitals?

Although in Epidemap the building information (e.g. hospital or school) is extracted from OpenStreetMaps, it is not taken into account in the present study for parsimony reasons. Indeed, our goal is to present a simple study without adding specific assumptions regarding the agents (e.g. depending of the kind of building where they encounter other agents). For instance, in Epidemap we could easily have mobility depend on age, e.g. move all the agent who are less than 18 years old to the closest school every day. We could also move every critical infected agent to the closest hospital. However, both of these require to make additional assumptions.

Regarding the way transmission takes place, it is important to stress that the time unit are slots of 8 hours in this study. During a specific time slot, an individual is either in her/his home building or in another building. If other individuals are in the same building during the time slot, a transmission event can occur. At the end of the day, individuals who are not in their home building are sent back there. Therefore, household and school/workplace encounters are very similar. The main difference is that individuals spend more time in their household than in any other building.

Regarding hospitals, transmission can occur there as it can in any other building (if more than one individual are visiting it during the same time slot). Modelling nosocaumial transmission could be interesting but, in the case of SARS-CoV-2, hospitalised patients tend not to be very contagious (more than 95% of the transmission events occur before day 11 and hospitalisation occurs on average on day 14). But for other types of infections where hospitalised patients are infectious, this could be particularly interesting.

We now better explain how the interactions occur in the Model section and we mention hospital transmission in the Discussion as a perspective for future work.

– The age of contacts is recorded, but it is not clear how/if this information is incorporated into the simulation; e.g. differences in disease severity profiles on the basis of age.

This point was also raised by Reviewer #1. Age is modelled but is currently only used for the hospital side of the simulations to stratify the infection fatality ratio. Therefore, age does not affect the spread of the infection. However, it would be interesting to also have household structure and differences in mobility with age.

We better explain how age is currently implemented and the role it plays in the simulation. In the Discussion, we present some model extensions to further explore the interaction between age stratification and spatial structure to investigate infection spread.

– How were the point estimates and 95% CI calculated?

The calculation was done using quantile method from the stats package in R. We selected type 8, the details and properties of which are fully described in Hyndman and Fan (Sample Quantiles in Statistical Packages, 1996), where it is also denoted as type 8.

We now explain in the Methods how the point estimates and 95% CI calculated.